# Microneedles Drug Delivery Systems for Treatment of Cancer: A Recent Update

**DOI:** 10.3390/pharmaceutics12111101

**Published:** 2020-11-17

**Authors:** Aravindram Attiguppe Seetharam, Hani Choudhry, Muhammed A. Bakhrebah, Wesam H. Abdulaal, Maram Suresh Gupta, Syed Mohd Danish Rizvi, Qamre Alam, Devegowda Vishakante Gowda, Afrasim Moin

**Affiliations:** 1Department of Pharmaceutics, JSS College of Pharmacy, JSS Academy of Higher Education and Research (JSSAHER), Sri Shivarathreeshwara Nagar, Mysore 570015, India; arvindas76@gmail.com (A.A.S.); ramsureshgupta@gmail.com (M.S.G.); 2Department of Biochemistry, Cancer Metabolism & Epigenetic Unit, Faculty of Science, Cancer & Mutagenesis Unit, King Fahd Medical Research Center, King Abdulaziz University, Jeddah 21589, Saudi Arabia; hchoudhry@kau.edu.sa (H.C.); whabdulaal@kau.edu.sa (W.H.A.); 3Life Science & Environment Research Institute, King Abdulaziz City for Science and Technology (KACST), Riyadh 11442, Saudi Arabia; mbakhrbh@kacst.edu.sa; 4Department of Pharmaceutics, College of Pharmacy, University of Hail, Hail 81481, Saudi Arabia; sm.danish@uoh.edu.sa; 5Medical Genomics Research Department, King Abdullah International Medical Research Center (KAIMRC), King Saud Bin Abdulaziz University for Health Sciences, King Abdulaziz Medical City, Ministry of National Guard Health Affairs, Riyadh 11426, Saudi Arabia; alamqa@ngha.med.sa; 6Department of Polymer Science and Technology, Sri Jayachamarajendra College of Engineering, Mysore 570016, India; siddaramaiah@yahoo.com

**Keywords:** transdermal drug delivery, non-invasive, breast cancer, skin carcinoma, regulatory

## Abstract

Microneedles (MNs) are tiny needle like structures used in drug delivery through layers of the skin. They are non-invasive and are associated with significantly less or no pain at the site of administration to the skin. MNs are excellent in delivering both small and large molecules to the subjects in need thereof. There exist several strategies for drug delivery using MNs, wherein each strategy has its pros and cons. Research in this domain lead to product development and commercialization for clinical use. Additionally, several MN-based products are undergoing clinical trials to evaluate its safety, efficacy, and tolerability. The present review begins by providing bird’s-eye view about the general characteristics of MNs followed by providing recent updates in the treatment of cancer using MNs. Particularly, we provide an overview of various aspects namely: anti-cancerous MNs that work based on sensor technology, MNs for treatment of breast cancer, skin carcinoma, prostate cancer, and MNs fabricated by additive manufacturing or 3 dimensional printing for treatment of cancer. Further, the review also provides limitations, safety concerns, and latest updates about the clinical trials on MNs for the treatment of cancer. Furthermore, we also provide a regulatory overview from the “United States Food and Drug Administration” about MNs.

## 1. Introduction

Drug delivery via the oral route is the most preferred route of drug administration due to its simplicity and cost-effectiveness when compared to other routes [1,2]. Nevertheless, this route suffers from disadvantages such as difficulty in swallowing [3], extensive first-pass metabolism leading to low bioavailability [4,5], long intestinal transit time [6,7], and poor or improper absorption [8]. On the other hand, the parenteral route of administration is a natural alternative preferred by the most to overcome the disadvantages set forth by the oral route. Traditionally, drugs are administered via parenteral route using a syringe and a hypodermic needle. Though the parenteral route was the natural (expensive) alternative, it did pose many complications to the patients like unbearable pain, thrombus formation at the site of administration, and hypersensitivity [9]. To overcome the disadvantages of oral and parenteral routes of administration, researchers focused on developing a novel drug delivery systems that help in delivering the drug(s) through the layers of skin, popularly called ‘Transdermal Drug Delivery System’ (TDDS) [10,11]. The dermis, a layer below the (epidermis) skin has decent vasculature that promotes drug absorption [12,13]. The main barrier for TDDS is the epidermis, “stratum corneum (SC)”, outmost layer of the skin. To make the drug travel across the SC, several methods/strategies were employed to improve drug delivery. Some of the methods/strategies include iontophoresis [14,15], concept of prodrugs [16], chemical enhancers [17,18], microsystems, and nanosystems [19]. The pros and cons of all these methods were recently published by Guillot and colleagues [20].

Cancer is a disease that is resulting due to changes in cellular level leading to uncontrollable growth and division of the cells. Its incidence is augmenting over the last few decades [21]. Standard treatments include but not limiting to chemotherapy, radiotherapy, and surgery. Nonetheless, these approaches are associated with low efficacy and severe side effects [22,23]. Besides, parenteral administration of anticancer drugs to the subjects often results in drug accumulation in different organs, low bioavailability due to rapid clearance from the blood circulation thus leading to low therapeutic efficacy [24,25]. In cancer treatment, the drug performance could be significantly enhanced by delivering it in the right quantity, at right time and at right site [26,27]. This can be achieved using Microneedles (MNs) [28], popularly considered as a synergistic combination of transdermal patch with hypodermic needles [29,30] wherein a drug is delivered directly to the tumor site via the micro-channels formed by MNs after piercing the SC of the skin [31,32,33]. Nonetheless, this is not true for all the types of cancer (except breast, prostate, cervical and skin) and all the therapies thereof. They are simple, non-invasive, and safer alternatives to the hypodermic needles and syringes. MNs are advantageous over hypodermic needles as they cause less or no pain at the site of administration and help in controlled and localized drug delivery with better flux [34,35]. Most importantly, site-specific delivery helps in preventing the death of healthy cells and thereby avoiding the side effects and maintaining the safety of subjects undergoing chemotherapy. Again, these aspects shall be considered by duly taking care of various challenges and limitations posed by MNs.

For the sake of clarity and brevity, the present review begins by providing general characteristics of MNs, wherein we highlight the information pertinent to conception of MNs, different approaches or strategies employed in drug delivery, the manufacturing methods, materials, clinical applications (with benefits, challenges and enhanced therapeutic outcome in clinical applications), design and geometry. The review continues by providing recent updates about the role of MNs as a carrier in delivering anti-cancer drug substances. While there are several developments in the domain of MNs in treatment of cancer, but this review is focused on discussing about sensor technology based MNs, MNs for treatment of cancer (breast, skin and prostate) and fabricating MNs by additive manufacturing or 3 dimensional printing (3DP). The review also provides limitations, safety concerns, and updates about the ongoing clinical trials of MNs for the treatment of cancer. Lastly, it discusses regulatory (USFDA) considerations on MNs.

## 2. General Characteristics of MNs

### 2.1. History

Based on our literature review, we found two US patents having inventors ‘Harvey Kravitz’ and ‘Norman Letvinn’ who were the first to conceive and reduce to practice the concept of MNs [36,37]. This evolving technology was reported as ‘means for vaccination’ or ‘vaccinating devices’. The objective of these inventors was to basically develop an inexpensive device for reliable delivery of vaccines by puncturing the layers of the skin. This concept was further developed with a suitable frame-work by the inventors Gastrel and Place of Alza Corporation in the year 1971 [38]. Figure 1 provides a pictorial representation of early MNs; the images are taken from respective patent citations by deleting the reference numerals. MNs just remained as a concept until 2000, with the exception of a journal article published by Henry and co-workers on microfabricated MNs [32]. The principles and the foundations set forth by early inventors (Kravitz and Letvinn) and the conceptual frame-work set by inventors of Alza Corporation are even followed today by many researchers in the domain of MNs.

### 2.2. Approaches for Drug Delivery

There exist different approaches/strategies for drug delivery using MNs [39], refer Figure 2. The popular strategies include poke and patch, coat and poke, poke and release and lastly poke and flow. In poke and patch, the MNs are pierced through the SC to create micro-channels. This is followed by removing the MNs and applying drug dosage form such as a cream/gel/solution, which acts as a drug reservoir, refer Figure 2a. One of the limitations of this approach is that the micro-channels close after a specific period of time. Thus, stopping the drug delivery [29,40,41,42,43]. In the second approach, coat and poke, the drug substance is coated over the MNs, refer Figure 2b. The steps involved in the process include piercing the SC using drug coated MNs, wherein the coated drug gets diffused from the coating layers into the layers of the skin. This approach is limited to drugs with very low dose. Additionally, drug coating decreases sharpness of the MNs [44,45,46,47].

Turning now to poke and release approach, wherein the MNs tend to dissolve after insertion, refer Figure 2c. Thus, eliminating the step of removal as seen with poke and patch strategy. Besides, the rapidly separating microneedles are proposed as a hybrid variety between coat and poke strategy and dissolving microneedles (as explained above). In this strategy, the drug loaded soft matrix remains in the skin while the solid is taken off. Further, hydrogel forming microneedles [48,49] interact with interstitial fluid and imbibe to form micro-channels for delivery of less potent drugs [50,51,52,53]. Lastly, the poke and flow strategy resembles the conventional hypodermic injection method with significantly less or no pain at the site of administration. Here the microneedles are loaded with drug solutions which upon insertion to skin releases the drug—refer Figure 2d, [54,55].

### 2.3. Manufacturing Methods and Materials

Turning now to MNs manufacturing methods, the most conventional methods of manufacturing include solvent casting, micro-molding, pulling pipettes, etching, lithography, ceramic sintering, laser cutting/ablation, electropolishing, and micro stereolithography [59]. Various organic and inorganic materials are employed in fabricating MNs of different structures and shapes [41], refer Figure 3.

As regards the materials employed in fabricating MNs, Guillot and colleagues identified various materials and methods used in the fabrication of MNs, they also provided the advantages and disadvantages of the respective methods [20]. For instance, polymeric MNs help in the delivery of molecules with large molecular weight such as DNA vaccine [53], nucleic acids, proteins [60] and others [61,62]. Similarly, Park and colleagues developed polymeric MNs for controlled drug delivery, wherein they encapsulated bovine serum albumin and calcein (a drug used in the treatment of lung cancer) on polymeric MNs [63]. RegeronInc, a Korean-based company, reported dissolvable MNs of heat shock protein (HSP90a and/or its fragments) which were prepared as per the methods described by Moga and colleagues [64,65]. Inventors Ronnander and Simon of LTS Lohman and New Jersey Institute of Technology developed a polyvinylpyrrolidone (PVP) MNs array patch for delivering macromolecular compounds such as sumatriptan or its succinate salt (used for migraine) by microporation of the skin. In addition, the PVP MNs array composition comprises glycerol that acts as a humectant/softener and polysorbate 80 acts as a surfactant. The patch of MN array helps in the delivery of said macromolecules using a current source that is controllable [66]. However, on a few occasions, MNs could not achieve an efficient therapeutic effect [67]. Such issues were studied and fixed by Wang and colleagues using polymeric MNs that are obtained using polyvinyl alcohol and maltose. Such a combination of polymers not only helped in achieving MNs with good mechanical strength but also MNs that can penetration through the layers of skin to generate micropores [68].

### 2.4. Clinical Applications

Based on our review, it appears that the clinical applications of MNs would not only depend on the ability of MNs array or patch to perform the intended function but also its acceptance across the board by the healthcare professionals. Several exploratory studies about end-user acceptance and views of healthcare professionals were reported in the past. Some of the benefits reported by the end-users about MNs technology include but not limiting to ease in administration (self-administration), less pain and tissue trauma for children. Nonetheless, few healthcare professionals raised concerns via-a-vis the dose, delayed onset, allergic reaction at the site of administration, and its potential for abuse by some subjects [69,70]. Furthering this, a follow-up study reported that the acceptability of hollow MNs for interstitial fluid and blood sampling for monitoring purpose has gained more acceptability than the conventional sampling with hypodermic needles. This group went on to suggest that the terminology of ‘needles’ shall be avoided as it creates a fear in the patient population [71]. Thus, it appears that MNs could play a pivotal role in gaining patient compliance in clinical sampling of body fluids for monitoring purposes in pediatric patient population.

#### 2.4.1. Clinical Benefits

One of the important clinical benefits of MNs is less or no pain at the site of insertion. For instance, Kaushik et al. carried out early trials to understand the painless insertion of MNs and reported that pain associated with MNs (400 MNs in 3 × 3 mm, height of the needle 150 µm) based drug delivery was statistically less when compared with that of the hypodermic needle [72]. Secondly, MNs are less invasive and form small lesion at the site of insertion when compared to traditional hypodermic injection used for systemic drug delivery. Nonetheless, some studies reported that drug delivery using MNs would lead to resolvable issues such as bleeding (minimum), oedema and erythema at the site of administration [73]. Another study reported that MNs fabricated using titanium showed minimum tissue trauma and faster recovery of the pores when compared to the ones induced by conventional hypodermic needle injections [74]. Thirdly, another advantage of MNs is the ease of administration as it does not require any healthcare professional to inject the drug substance as seen with hypodermic needle-based injections. For instance, Jin et al. proposed MNs of chitin coated with tuberculosis (TB) antigen as an alternate to the Mantoux test, wherein a purified TB antigen is injected intradermally at an angle of 5–15 using a hypodermic needle. It is pertinent to state that any variation to the style of administration leads to failure of the test. The chitin coated MNs were tested in guinea pigs that showed positive BCG test against TB [75,76,77]. Therefore, it appears that MNs could help in preventing administration errors to an extent and also helps in gaining subjects compliance by allowing self-administration. Despite various clinical benefits of MNs, there are still many challenges that one needs to fix before they are set to make any clinical impact. Some of the challenges/limitations are discussed in the below section.

#### 2.4.2. Clinical Challenges

Insertion of MNs on the surface of skin and formation of pores pose a threat of microbial infection leading to local or systemic infections. Therefore, traversing the pores generated post insertion of MNs on the surface of skin is vital and this aspect was studied by Donnelly and colleagues using porcine skin and reported that pores formed due to MNs are less permeable to microbial penetration when compared with pores formed by hypodermic needle injections [78]. Nonetheless, the limitations of this study included the limited length (280 µm) of MNs and lack of simulating the exact skin microbiota for microbial permeation studies [79]. The risk of microbial infection could be mitigated by aseptic fabrication methods [80] or by employing microbial resistant polymers that were proposed by Hook et al. [81]. Such polymers prevent microbial adhesion and also penetration into the layers of skin when MNs are inserted.

Another risk in employing MNs based drug delivery system is skin irritation, which is generally transient in nature. Even repeated application of MNs is found to be safe [82]. Yet another major limitation is drug delivery using MNs is the amount of drug that is be delivered. For instance, in solid MNs, the time duration of the pores to remain open is a challenge as the skin tends to undergo natural regeneration process. This acts as a major impediment/limiting factor for drug delivery using solid MNs. Nonetheless, strategies to prolong the pore viability using diclofenac (COX inhibitor) and fluvastatin (HMG-CoA reductase) [83,84].

#### 2.4.3. Clinical Applications with Improved Therapeutic Outcome

The above aspect is explained using ‘Psoriasis’—a skin disease. It is a type of skin disease wherein the subject develops red, itchy, and scaly patches [85]. Existing treatment modalities provide symptomatic relief with no permanent cure. Besides, topical therapies were found to be time-consuming with various side effects namely irritation at the site which acts as an impediment to gain patient compliance. Additionally, systemic exposure of methotrexate has several adverse effects such as thrombocytopenia, nausea, vomiting, and hepatotoxicity [86,87]. Therefore, attempts were made to deliver methotrexate to the layers of skin directly at psoriatic sites. For instance, Vemulapalli and colleagues studied transdermal delivery of methotrexate using solid MNs of maltose using iontophoresis method. The combination of MNs with iontophoresis resulted in the 25 fold increase in delivery of methotrexate when compared with individual modality of administration [88]. In another study, Gujjar and colleagues reported successful transdermal delivery of pentaerythritol tetrakis (mate tea extract) using MNs. The method involved pre-treatment of subject’s skin with MNs followed by applying the emulsion (oil in water) formulation of the extract [89] which had poor solubility and was selectively portioning towards the SC of the skin.

### 2.5. Design and Geometry

Design and geometry (shape, tip radius and length, base width, aspect ratio, etc., of MNs play an important role not only in deciding the mechanical features but also the overall performance. Therefore, one has to judiciously select a right material and right methodology to fabricate ideal MNs that are biocompatible, mechanically strong to pierce layers of the skin, capable of loading different active drug substances and also controlled or sustained drug delivery [90]. Wang and Xu carried an excellent review on the aspect of different polymeric materials and methods employed in fabricating and obtaining ideal MNs [69]. The most vital aspect that is geometry of MNs facilitates smooth insertion onto the skin. This is important because human skin is elastic and robust in nature and thereby it tends to prevent MNs from insertion or may even lead to breakage of MNs. This is seen with MNs which are fabricated using weak materials with blunt tips [28,91]. Therefore, the geometry of MNs is sensitive and crucial for efficient drug delivery.

Different shapes of MNs that were developed include but not limiting to rectangular with sharp edge, cylindrical, pointed, pyramidal, conical, and octagonal with varying lengths and widths, Figure 4 provides few shapes of MNs. For instance, Lee et al. reported polymeric MNs that are of pyramidal shape with good mechanical strength vis-a-vis the conical shaped MNs as the pyramidal ones are associated with higher cross-sectional area at the same width of the base [92]. Additionally, Chen and colleagues reported that higher insertion depths were achieved with chitosan MNs having a tip radius of 5 µm when compared with that of the 10 µm tip radius MNs [93]. Similarly, the sharpness of the tip of MNs is vital for insertion to skin as MNs with sharp tip possess higher potential to penetrate with less force and the vice-versa is true with tips of MNs that have a larger diameter [31,94]. Ideal MNs for clinical application must be biocompatible, mechanically strong with good drug cargo loading potential. Such ideal MNs could be employed in transdermal drug delivery, bio-fluid extraction for diagnostic purposes and for transdermal sensing to analyze the concentration of different analytes in the interstitial fluid [95,96,97]. Refer to Figure 4 for design and geometry of MNs. All in all, it appears that pyramidal shaped MNs have sharper tips with low aspect ratio are associated with excellent skin insertion properties.

## 3. Different Cancer Treatments Using MNs

MNs help in overcoming the limitations associated with traditional drug delivery systems. They have been successfully evaluated for delivery of small molecules [98], large molecules [99,100] like chemotherapeutic agents, proteins, and genetic material. MNs also help in delivering microparticles or nanoparticles of anticancerous drug substances [101]. In the following sections we discuss MNs that work based on sensor technology, MNs that are used for treatment of breast cancer, skin carcinoma, and finally the 3D printed MNs for treatment of cancer.

### 3.1. Sensor Technology Based MNs

Integration of MNs with microelectronic sensors is an emerging domain. The sensor based MNs have the potential to convert non-electrical inputs like temperature or pressure from its surrounding environment to microcomputer readable electrical signals [102]. Traditionally, blood samples are used/withdrawn from subjects to measure the concentration of the analyte. Currently, ‘point-sample’ approach is used, wherein blood is collected as a drop onto a test strip followed by inserting into an electronic device to display the result on the display screen. The existing methods suffer from various limitations such as pain, risk of infection/cross-contamination due to multiple uses, slow skin healing after collecting the sample. One alternate to overcome the above limitation is usage of sensor based MNs, which cause no or less pain compared to hypodermic needles, non-invasive, safe, and easy to use. Generally speaking, there are four different types of microneedle-based sensors and their principles of working are illustrated in Figure 5.

The sensors for MNs basically comprise of analyte recognition element and a transducer (electrochemical or optical). As regards the colorimetric MNs sensor, the enzyme glucose oxidase is used for analyte recognition and catalysis to release hydrogen peroxide. The hydrogen peroxide induced color change of a substrate is used to determine the concentration of the analyte [103]. Similarly, in immunosensor MNs, enzyme-linked immune sorbent assay is used. Hydrogel MNs are employed to immobilize nucleic acids to facilitate binding of complementary DNAs based on Watson and Crick base pairing [104].

Some of the MNs that work based on sensor technology are discussed below and the same are illustrated in Figure 6.

In one of the studies, wearable orthogonal MNs that work based on sensor technology were successfully developed by Goud and colleagues for monitoring levodopa levels in the interstitial fluid of patients with Parkinson’s disease. This helps in proper optimization of the dose of the drug, levodopa [105]. Hahn and colleagues from Postech Academy-Industry Foundation, Korea reported MNs that work on sensor technology. The entire MN device has the potential to sense the presence of nitrogen monoxide (innate immune response to cancer) using electrochemical doctrine. This device was helping to detect cancer, the growth, and the size of a tumor. The proposed MNs were associated with a polymeric base that has layers of adhesive polymer (chitosan, fibronectin, vitronectin, polydopamine, silk, and collagen), conductive polymer (polyacetylene, polypyrrole, polyphenylene vinylene, and others) nitrogen monoxide bonding molecule with iron ions (porphyrin or heme) as a layer and electrodes (working and counter electrodes) such as nickel, gold, silver for detecting nitrogen monoxide. The patches proposed by Hahn and colleagues have the potential for usage in various cancer treatments namely breast, lung, uterine and colorectal cancers [106], refer Figure 3a—adapted from the respective US patent.

Turning now to other sensing mechanisms, ‘The Regents of the University of California’, North Carolina State University along with Sandia Corporation developed MNs that works based on biosensing mechanism to deliver the drug. The disclosed MNs are pictorially depicted in Figure 3b, (adapted from respective reference) wherein the MNs make use of probes that are used for sensing the signal and which in turn are connected to electrically conducting wires for transmitting the detected signal generated by the probes. Interestingly, these MNs could be used for the simultaneous detection of “glucose, lactic acid, and pH”. These parameters are important to determine malignant skin cells because in cancer the lactate levels increase while glucose levels and pH decrease. Therefore, the proposed MNs help in determining malignant skin cells in the body [107]. In another study, Rylander and colleagues disclosed fiber optic-based MN device that has the potential to not only penetrate the tissue but also deliver light at the required area of the layers of the skin, refer Figure 3c (adapted from respective reference). The device delivers light-based therapeutic agents on the layers of skin for the treatment of cancer and is also used for photothermal/photochemical therapy for the subjects who are in need thereof [108].

For instance, Jain and team reported dip-coated MNs (57 in number) of 5-Aminolevulinic acid (5-ALA) for ‘photodynamic therapy’ (PDT) of various skin related carcinoma [109]. Donnelly and colleagues were the first to use silicon-based solid MNs of 5-aminolevulinic acid (ALA), a skin photosensitizer. They tested the MNs in a mouse model using poke and patch strategy. The study revealed that a reduced dose is sufficient to elicit high levels of ALA in skin lesions [110]. In yet another study, Zhao et al. disclosed fast dissolving tip-loaded microneedle patches of 5-ALA (122 µg per tip of the needle) that are fabricated using sodium hyaluronate as a polymer for subcutaneous tumor therapy. The prepared MNs were found to be safe with good mechanical strength to pierce (200 µm depth) the SC followed by dissolution and release of the drug. The study concluded that tip-loaded MNs of 5-ALA showed better tumor inhibitory rate (97%) when compared with injection formulation (66%).

5-ALA has an inherent natural property wherein it converts to a ‘photosensitizer called protoporphyrin IX (PPIX)’, refer Figure 3c, inside the cells through various transformations in the mitochondria, which undergoes chelation with iron to produce heme in the presence of an enzyme ferrochelatase [111].

Combination cancer therapy (photothermal and chemotherapy) also helps in achieving desired efficacy in treatment or eradication of cancer [112]. Polyvinylpyrrolidone MNs coated with chitosan and polyvinyl alcohol to arbitrate the delivery of doxorubicin nanoparticles [113]. Bhatnagar and team reported MNs of polyvinylalcohol (PVA) and polyvinylpyrrolidone (PVP) that are dissolving type for delivery of doxorubicin and docetaxel for breast cancer treatment [114]. Similarly, Hao and team employed synergistic tumor therapy (PTT with chemotherapy) using near-IR responsive docetaxel loaded gold nanorod coated polymer poly(l-lactide) microneedles for treatment of A431 tumor. The MN system showed promising results with no recurrence. Nonetheless, it suffered from multiple intravenous administrations [115]. To overcome this limitation, Hao and colleagues developed a “PEGylated gold nanorod and doxorubicin-loaded dissolvable hyaluronic acid microneedle system”. This system exhibited good effects (photothermal and transfer of heat) and treatment efficacy. Nonetheless, this was associated with a disadvantage of burning the skin at the site of administration. Additionally, the proposed MN system was exhibiting abrupt drug release [116]. In order to overcome the above limitations, very recently, Hao and colleagues developed a controlled release, dissolvable and single-dose MNs of 5-fluorouracil (near-IR light-responsive) and indocyanine green (photothermal agent approved by USFDA) polymer (monomethoxy-poly (ethylene glycol)-polycaprolactone (MPEG-PCL) loaded nanoparticles integrated with USFDA approved hyaluronic acid for treatment of skin cancer in general and particularly human epidermoid cancer and melanoma. The release of the drug from the nanoparticles is controlled by near-IR light to a achieve single-dose cure for skin cancer.

From the above discussion, it can be inferred that different analytes could be targeted for microneedle-based cancer detection. For instance, the target analytes include but not limiting to glucose, glutamate, lactate, hydrogen peroxide, nitric oxide, potassium levels, and T cell. Additionally, determining the pH of the tumor environment is also important in cancer detection using MNs. A table depicting different types of analytes relied on detecting cancer, the principle involved in the detection and the test subjects on which the tests were carried out are provided in Table 1.

### 3.2. MNs in the Treatment of Breast Cancer

After skin cancer, the second and most common cancer that causes death in women is breast cancer [129]. Some of the therapies include chemotherapy to prevent the recurrence of the tumor or surgical resection to remove the local tumors [130,131]. Generally, chemotherapy involves systemic administration of the anti-cancerous drugs that often lead to various adverse effects such as suppression of bone marrow, cardiac and/or neurotoxicity [132], mastectomy, hormone therapy, lumpectomy, radiation therapy, and biological therapy using macromolecules [133]. In search of a better line of treatment oncologists shifted their attention towards immunotherapy and cancer vaccines. Nonetheless, all the traditional therapies for breast cancer treatment suffered with toxicity and less therapeutic efficacy.

As regards the biology, breast cancer originates in the ducts of mammary glands that help in carrying milk from the lobules to the nipple. Drug delivery to the underlying tissues (tumors) in the breast helps in minimizing its blood concentration and thereby protecting the healthy cells and adverse effects [134]. The key site of breast cancer is the nipple duct which extends further as a lobular duct. The ‘Nipple-areola’ complex helps in enhancing the drug permeation as the epidermis is thinner in this region when compared to SC of the skin. Further, this region also shows more appendages of sebaceous gland, sweat gland, apocrine, and eccrine glands that act as alternate transport paths when compared to the primary paths, namely the transepidermal and transductal routs [135]. Researchers investigated that localized drug delivery through the mammary papilla helps in significantly inhibiting the growth of cancer cells as the drug gets localized to the tumor site [136]. Localized drug delivery using MNs is set to disrupt the way the drugs are delivered to the tumor site [137,138]. Some of the MNs based anti-cancerous drugs delivered for treatment of breast cancer are discussed below:

Bhatnagar and colleagues reported MNs of zein (prolamine protein from corn) for delivery of tamoxifen and gemcitabine that are used in the treatment of breast cancer. Zein MNs were prepared by micromolding technique through solvent casting, wherein it involved 3D printing of master mold using a polymer, acrylobutyl nitrile styrene, and plasticizers namely glycerol and PEG 400. Both tamoxifen and gemcitabine were encapsulated in zein MNs. Alternately, both the anti-breast cancer drugs were also coated on zein MNs using rhodamine as a dye to examine the quality of coatings. The studies on loading efficiency revealed that more tamoxifen was loaded onto zein MNs when compared to gemcitabine. As regards the release kinetic studies, gemcitabine showed more permeation vis-a-vis tamoxifen. Finally, the study concluded that optimal solubility of a drug in water may exhibit greater permeation when delivered using zein MNs [139]. In another study of Bhatnagar and colleagues, they reported dissolving MNs prepared using polyvinyl pyrrolidone and polyvinyl alcohol for co-delivery of doxorubicin and docetaxel for treatment of breast cancer. The MNs were prepared using micromolding technique as previously explained [140], drugs (doxorubicin and docetaxel) were loaded by encapsulation method using polyvinyl pyrrolidone (PVPK360) and the base plate was prepared using both PVPK360 and polyvinyl alcohol. The prepared MNs dissolved in less than an hour after inserting it into the skin. They employed 4T1 breast cancer cells to determine the effectiveness of MNs. The toxicity studies in animals showed that the administration of both the drugs using MNs showed a higher survival rate compared to that of injections (intratumoral). Co-delivery of these two drugs showed controlled growth of the tumor than when they were administered individually [114].

Gao and co-workers reported transdermal delivery of a biphenyl natural anti-cancer compound, “*Honokiol*”, refer Figure 7, obtained from different plant parts of *Magnolia glandiflora*.

Reports also suggest that *Magnolia glandiflora* is indeed associated with various anti-cancer properties; breast [141] prostate [142] liver [143]. MNs made of maltose were used in delivering honokiol through the layers of skin and also through the mammary papilla. In addition to MNs based delivery of honokiol, Gao and co-workers also studied the delivery of this natural anti-cancer compound using skin permeation enhancers. The study used chemical enhancers such as propylene glycol, oleic acid and isopropyl myristate. Oleic acid was found to be an excellent penetration enhancer when compared to other agents. Both the delivery systems (MNs and penetration enhancers) were found to be promising and progressive in delivering honokiol through the layers of skin. Additionally, the anti-cancer effect of honokiol was evident with decreased release of cytokine interleukin-6 and suppression of Ki-67 protein [144].

Chablani and colleagues used metallic MNs (AdminPatch^®^ 1200) to deliver particulate (1.5 µm size) breast cancer vaccine prepared by freeze-drying method. They used whole-cell lysate as a source of antigens. The micro channels (50 ± 10 µm) formed using MNs were microscopically visualized using methylene blue as a staining agent. Based on the in vivo studies, it is evident that the overall approach was found to be a very promising means of immunization against breast cancer [129]. Mojeiko and the team reported the usage of MNs to augment the penetration of celecoxib microemulsion in the treatment of breast cancer. MN-based roller treatment in combination with microemulsion containing 60% of water was found to significantly enhance celecoxib penetration through the layers of mammary tissue [145]. Additionally, MNs could be considered for drug delivery via the transpapillary route (breast) for the treatment of breast cancer [146].

Chen and team employed MNs having a nano-silver/MBL membrane that works on the siphoning effect for early detection of breast cancer. The method employs the insertion of MNs at the desired area of the breast thereby opening the interstitial space to collect the sample/cells. Due to the siphoning effect of the film the cells get absorbed onto the membrane which is subjected to colorimetric analysis to detect early stages of breast cancer in subjects [147]. Zandi and co-workers reported MNs of electrochemical probes tinted by zinc-oxide nanostructures were used to generate microbubbles. After insertion of such MNs into the tumor interstitial fluid, sonoporation was employed for the formation of microbubbles by electrolysis to deliver paclitaxel [148].

### 3.3. MNs in Skin Carcinoma

Skin carcinoma is the most common type of cancer and reports indicate that one in five individuals in America would be affected in their lifetime [149]. Different types of skin cancer are basal cell carcinoma, squamous cell carcinoma and melanoma carcinoma [150]. The first two are more common when compared to melanoma carcinoma; a more dangerous when compared to the other two. Skin cancer is mainly caused due to long exposure to UV radiations. Traditional therapies such as chemotherapy, radiotherapy, surgical resection, and immunotherapy are associated with various limitations [151,152,153,154]. Besides, anti-cancerous drugs when administered via oral or parenteral route undergo gastric degradation or first-pass effect thus leading to various side effects. MNs appear to be the safest alternatives in view of the aforesaid limitations. This is because MNs have the ability to penetrate the SC and deliver the drug in the dermis region via the acanthocyte and basal cell layers [155,156]. In light of the biology of the skin carcinoma drug delivery via the MNs is the best for treatment of skin carcinoma [60,157]. Some of the MNs based anti-cancerous drugs delivered for treatment of skin carcinoma is discussed below:

North Carolina State University inventors Zhen Gu and others reported self-degradable and sustained-release MNs patch of PD1 antibodies (nivolumab, and pembrolizumab) either alone or in combination with anti-CTLA4 antibody (ipilimumab). The proposed MN patch comprises a base end and a tip coated with nanoparticles that encapsulate the immunotherapeutic agent (PD1 antibody and/or anti-CTLA4 antibody) along with glucose oxidase as a pH altering agent. The nanoparticles prepared were acid-degradable and are prepared using dextran and sodium alginate as a surfactant. In some embodiments, Zhen Gu and colleagues used hyaluronic acid in the preparation of MN patches, which helps in lubricating the layers of the skin when the patch is inserted [158,159]. Zynerba Pharmaceuticals, a US company, disclosed MNs comprising cannabidiol and its prodrugs that have the potential for usage in pancreatic cancer. It proposes a hydrogel, a reservoir type patch of MNs [160].

Eirion therapeutics filed two international patent applications on transdermal delivery of large molecular weight molecules that are used in the treatment of skin cancer. The invention claimed a method for delivering an emulsion formulation of botulinum toxin (molecular weight of 100,000 KDa) with a biologically active agent (hydrocortisone, lidocaine, retin A, and others). The method involved steps of skin conditioning using MNs followed by the application of the formulation. They have basically combined the microneedle technology with emulsion technology (water-in-oil and oil-in-water nanoemulsions) to deliver botulinum toxin, a large molecular agent. The emulsion formulation also comprised non-irritating and penetration enhancing agents such as “cationic peptide” and a positively charged carrier having the sequence “RKKRRQRRRG-(K)_15-_GRKKRRQRRR”. The invention also disclosed the use of high molecular weight biologically active agents such as infliximab, golimumab, adalimumab, certolizumabpegol, siplizumab, and others either alone or in combination with one another [161,162].

Lu and co-workers reported dacarbazine loaded MNs for skin carcinoma. The MNs were prepared by micro stereolithography using poly (propylene Fumarate) as a polymer and diethyl fumarate was added to control the viscosity of the polymer. The prepared MNs have a cylindrical base and a conical tip with excellent skin insertion force. Dacarbazine was released in a controlled fashion for the above five weeks [163]. Hamdan and colleagues reported dissolving MNs of “5,10,15,20-tetrakis(2,6-difluoro-3-*N*-methylsulfamoylphenyl bacteriochlorin”, a preformed photosensitizer (near-IR). The studies revealed that the MNs had good mechanical strength for insertion into the layers of the skin, 5mm down the skin for treatment of nodular skin carcinoma. Thus, the study suggested usage of polymeric (co-polymer of methyl vinyl ether and maleic acid) MNs in photodynamic therapy of skin lesions that are deep [164]. Al-mayahy and team reported the usage of MNs to enhance penetration of imiquimod for the treatment of basal cell carcinoma [165]. In another study on imiquimod, Sabri and colleagues reported usage of oscillating MNs pen ‘Dermapen’ and ‘Dermastamp’ to enhance the penetration of imiquimod when applied as a cream using ‘poke-and-patch’ and ‘patch-and-poke’ strategies. Release kinetic studies confirmed that imiquimod penetration was significantly higher when administered by the ‘patch-and-poke’ strategy using ‘Dermapen’. The prime reason behind the same was due to the application of MNs leading to the formation of the intra-dermal depot of drug and isostearic acid (cream excipient) that lasts for about a day [166].

Inventors from the University of Pittsburgh and Carnegie Mellon University disclosed carboxymethyl cellulose (CMC) MNs that can deliver various chemotherapeutic and immune-stimulating agents (Toll-like Receptor (TLR) antagonist) either alone or in combination. For instance, the chemotherapeutic agent was selected from a group comprising of doxorubicin, valrubicin, epirubicin, idarubicin, and other known anthracycline agents for treatment of skin cancer. The MNs were prepared using CMC, wherein the drugs are deposited by micro milling method in the form of drug cargo layers, as shown in Figure 8, (adapted from respective reference) [167]. Jiangsu college of information technology disclosed semiconductor-based microneedles that help not only for gene delivery but also for the delivery of anticancerous drugs [168].

### 3.4. MNs for Prostate Cancer

The second most common cancer in men after lung cancer is Prostate Cancer (PC) [169]. It is commonly seen in male population of the developed countries and the mortality rate is just behind lung and colorectal cancer. Existing treatment modalities for PC include but not limiting to radical prostatectomy (surgical treatment for low risk PC), radiotherapy or image-guided radiotherapy, Androgen Deprivation Therapy (ADT) either alone or in combination with radiotherapy and brachytherapy (radioactive material is positioned inside the body). The castration resistance seen in ADT is treated using various drugs namely abiraterone acetate (CYP17-α-hydoxylase inhibitor), docetaxel, prednisone, cabazitaxel and enzalutamide [170].

Anti-cancer vaccines, such as PC DNA vaccines, are other alternatives that can be employed to target cells that are expressing immunogenic tumor-associated antigens (TAAs). Nonetheless, recent clinical trial results on PROSTVAC raised concerns about the efficacy of DNA vaccine alone [171]. The effectiveness of a PC DNA vaccine could be enhanced by selecting an appropriate drug delivery system such as MNs. The beauty of MNs is that the drug can be delivered locally to the tumor site via the layers of the skin to the Antigen Presenting Cells (APCs) in the epidermis and dermis regions [172].

For ADT, Chen and colleagues reported transdermal delivery of goserelin (luteinizing hormone-releasing hormone (LHRH) analogue) using dissolvable polymeric MNs of Chitosan and polyvinyl alcohol/polyvinyl pyrrolidone base as a support. The LHRH per each microneedle was 73.3 ± 2.8 µg [173]. The pores formed after piercing the skin would reseal after 7 days and the serum LHRH levels augmented initially followed by a decline at the end of the 7th day. As regards the testosterone levels, they increased to peak by 14th day and declined to the level of castration by end of 21st day. Additionally, electrically functional MNs were reported for human PC cells in vitro by electroporation method. Choi and colleagues reported MNs fabricated by micromolding technology to electroporate human PC cells for enhancing transfection by DNA vaccines [174,175].

Very recently, Cole and colleagues reported for the first time a robust two-tier microneedle-based platform for PC DNA vaccine delivery. In its study, RALA/pDNA nanoparticle based dissolving polymeric microneedle system was developed to immunize mice encoded with mPSCA (Prostate Stem Cell Antigen). The results of the study showed production of encoded PSCA. Additionally, immunization using RALA/pPSCA based MNs exhibited anti-tumor activity (inhibited TRAMP-C1 tumors) in the PC models [176].

## 4. More Recent Method of Manufacturing MNs—Printing Method

The traditional manufacturing methods and materials employed in fabrication of MNs pose different challenges that hamper its adoption for TDDS. For instance, the micro-molding method used for fabricating gel-forming and dissolving MNs is a time consuming and tedious process that often poses challenges with respect production scale-up. Additionally, the MNs drug coating technologies suffer with non-uniformity, lack of accuracy and uncertainty about the amount of drug that is loaded or coated on the MNs. Further, the materials used in fabrication of MNs are basically approved for oral use but majority of them are being utilized to fabricate MNs for drug delivery via the transdermal route. This leads to the problem of biocompatibility leading to irritation at the site of administration. One alternate and upcoming method of manufacturing MNs is additive manufacturing or 3-dimensional printing (3DP). The first three dimensional printed MNs for wound healing were reported by Gittard and colleagues, wherein they employed 3DP technique called steriolithography (SLA) to fabricate MNs followed by coating them with zinc and silver oxide [177]. In another study, Lu and colleagues reported dacarbazine printed MNs for treatment of skin cancer using micro SLA technique [163]. Some of the recently reported printed MNs are discussed below:

Uddin and colleagues reported coated MNs of 5-fluorouracil, curcumin, and cisplatin, prepared by inkjet printing technology. The formulation used for coating comprised of a drug-polymer at different ratios and Soluplus^®^ was used as a hydrophilic graft copolymer [178]. In another study, Uddin and colleagues also reported 3D printed MNs of Cisplatin to “A-431 epidermoid skin tumors”. The 3D printed MNs were prepared by the stereolithography method and cisplatin was coated using inkjet dispensing on the MNs surface. The drug release of about 80 to 90% was achieved in the first 1 h with good permeability when evaluated using Balb/c nude mice. The tumor inhibitory effect was evident through histopathology studies [179].

In short, employing 3DP technologies in fabricating MNs has a strong potential in printing complex MNs that need to be organized in a systematic fashion. The beauty with this technique is that the product is printed layer-by-layer [101]. Such printed MNs could help in not only enhancing the delivery but also the efficacy of anti-cancer drugs.

## 5. Clinical Trials (CTs) on MNs of Anti-Cancerous Drugs

Many CTs on MN-based drug delivery systems are carried out across the globe. The present review is focused on providing details about MN-based drugs that are used in the treatment of cancer. Reported below are studies that are recruiting patients/subjects and active studies but not recruiting subjects.

### 5.1. Recruiting

The University of Pittsburgh is recruiting subjects for its ongoing study to test the safety and effectiveness of microneedle applicator of doxorubicin, a chemotherapeutic agent [180]. Doxorubicin kills the tumor cells and alters its microenvironment, thus inducing systemic immunity at the tumor-specific site. The study aims to determine the best dose doxorubicin that could be used in the microneedle patch [181]. Additionally, Skinject Inc., a US company is recruiting subjects for its phase 1 study on basal cell carcinoma. The study is aimed to assess the dose (tolerated or toxicity), safety, efficacy, and tolerability of dissolving microneedle arrays loaded with doxorubicin. Skinject Inc. is carrying out these clinical studies at ‘The Center for Clinical and Cosmetic Research’, Florida, USA [180].

### 5.2. Active, Not Recruiting

Pulse biosciences, a US based company has an ongoing active clinical study on its proprietary platform technology branded as Nano-Pulse Simulation^TM^. It makes use of electrically conducting microneedles to apply nano electric pulses at targeted tissue and is aimed to study the effect in benign skin tumors [182,183]. Nonetheless, this study is currently not recruiting or enrolling potential subjects [180].

## 6. Limitations and Safety Concerns of MNs in Treatment of Cancer

Despite significant research in the domain of MNs, it still suffers from different challenges that are to be considered while deciding to use them as carriers for anti-cancer drug delivery. They are preferred for anti-cancer drugs that have low dose. Nonetheless, hydrogel-forming and hollow MNs could be used to deliver small or large molecules with high dose MNs that are to be used repeatedly are generally restricted to avoid safety concerns [49,184]. The drug delivery schedule and sequential release of anti-cancer drugs is still a challenge [185].

Generally speaking, depending on the duration of use of MNs they are associated with three safety concerns namely pain at the site of administration when used repeatedly for a long time, infection and reaction of the skin after inserting MNs. Pain depends on the duration of MNs use and also the number of needles in the MNs patch. For instance, Gill et al. reported less pain than the pain caused by a hypodermic needle with a gauge of 26, this study also provided correlation between the level of pain and number of needles and its length [186,187].

Infections could be due to entry of microorganisms via the micro-channels after insertion of MNs. The probability of such entry is less but it cannot be ruled out. Therefore, administration of MNs shall be done carefully to the subject to prevent such infections. Additionally, the length of the needle also plays a pivotal role in preventing/controlling the infections, if any, after inserting MNs [188].

Skin reactions such as erythema (redness of skin), inflammation, and irritation at the site of administration are observed with MNs made of materials such as glass and ceramics [34,189,190]. Additionally, MNs fabricated using steel may get corroded over a period of time and the same could be prevented by replacing steel with titanium [191].

## 7. Conclusions

MNs are emerging drug delivery systems that are associated with unique advantages as depicted in Figure 9. They have made a tremendous progress across all the areas of application namely diagnosis of a disease, treatment, immune-biological, dermatology, cosmetic applications, and treatment of cancer. Additionally, developing a wearable and smart sensor-based device for long-term treatment, personalized diagnostics and wireless transformation of the data are the areas yet to be explored. As an emerging drug delivery device, they can be successfully employed in not only delivering small or large molecules but also nanoparticles of anti-cancer drug substances. Selecting suitable material, design, geometry, and method of manufacturing MNs plays a crucial role in gaining patient compliance and drug release. Most importantly, fabricating MNs by 3DP is gaining popularity as it is simple and straightforward method that can help in printing complex MNs device layer by layer with appreciable accuracy. It appears that MN-based anti-cancer drug delivery systems are gaining popularity in treatment of different cancer types (skin, breast, prostate, etc.). Nonetheless, more research is required in this domain to ascertain the employability of MNs for treatment of different cancer types that are inherently inaccessible to the patch of MNs when positioned on the surface of the skin. Once more understanding is gained towards the existing challenges/limitations of MNs one could successfully translate them into clinical application. Last but not least, strong research collaborations between industry, academia, and research scientists help in translating and transitioning from laboratory setting to clinical practice for treating subjects in need thereof.

## Figures and Tables

**Figure 1 pharmaceutics-12-01101-f001:**
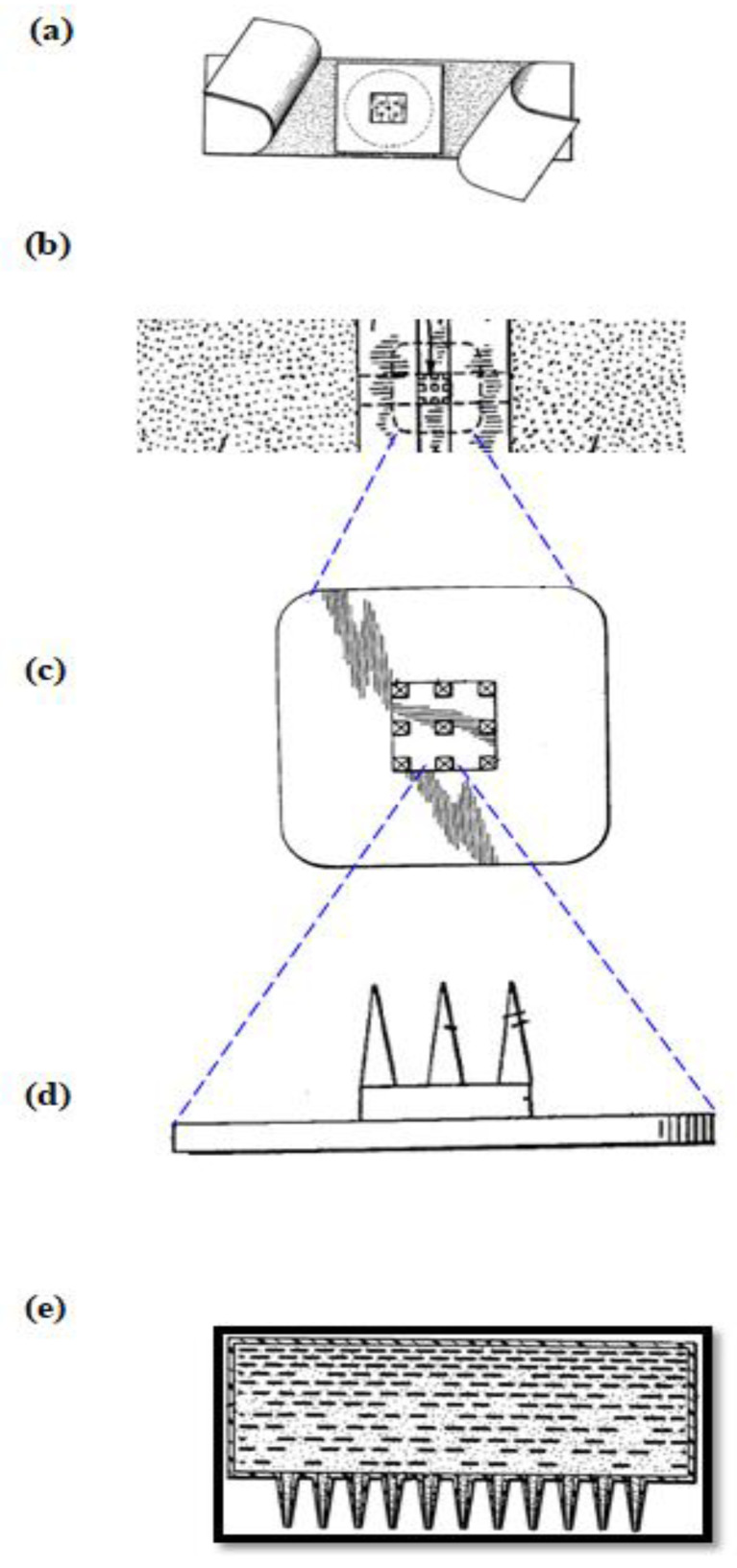
History of Microneedles (MNs) (**a**) Plan view of the microneedle device from reference [36] (**b**–**d**) Plan view of microneedle device, substantially to scale, blown-up to shown the MNs from reference [37] (**e**) MNs patented by Alza Corporation, from reference [38].

**Figure 2 pharmaceutics-12-01101-f002:**
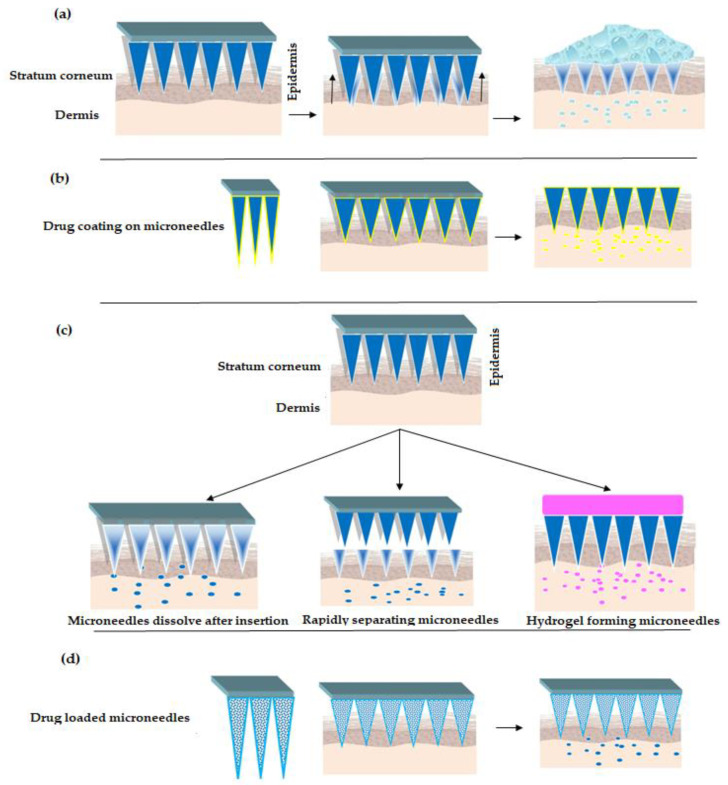
Strategies for drug delivery using MNs (**a**) “Poke and Patch” Strategy (**b**) “Coat and Poke” Strategy (**c**) “Poke and Release” Strategy [30,56,57,58] (**d**) “Poke and Flow” Strategy.

**Figure 3 pharmaceutics-12-01101-f003:**
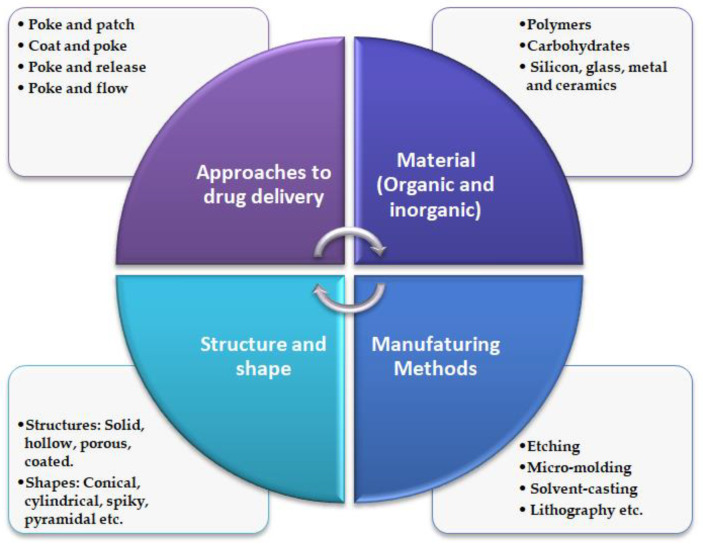
Schematic illustration of general characteristics of MNs.

**Figure 4 pharmaceutics-12-01101-f004:**
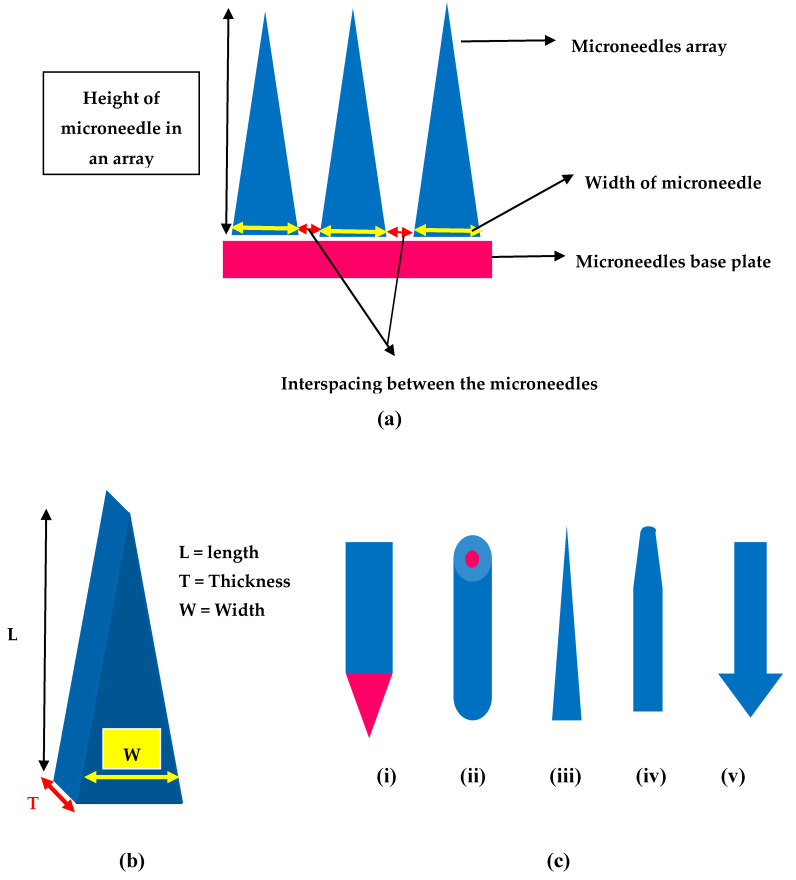
Design and geometry of MNs (**a**) Illustration of geometrical parameters in a microneedle array (**b**) Geometry of MNs length (L), thickness (T) and width (W) (**c**) Different shapes of MNs (**i**) Rectangular microneedle with sharp edge (**ii**) cylindrical (**iii**) conical (**iv**) tapered (**v**) arrow headed.

**Figure 5 pharmaceutics-12-01101-f005:**
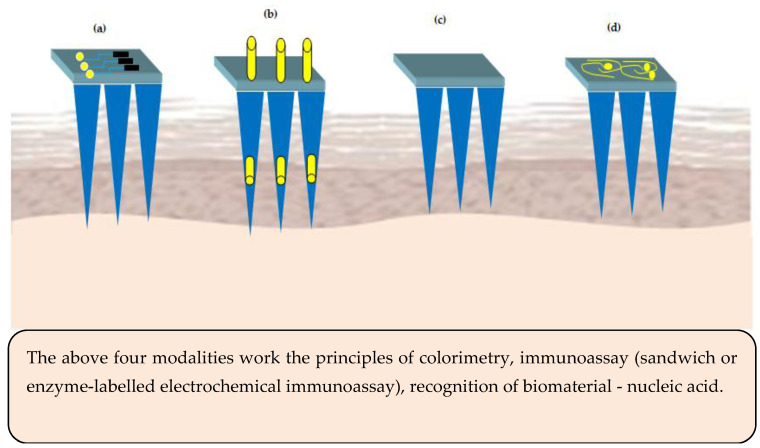
Microneedles (MNs) sensing modalities (**a**) sensor positioned on the MNs base or support (**b**) electrode inserted into the hollow microneedles—acts electrochemically (**c**) surface of the MNs is functionalized to act as a sensor (**d**) MNs are metallized to act as bio-electrodes.

**Figure 6 pharmaceutics-12-01101-f006:**
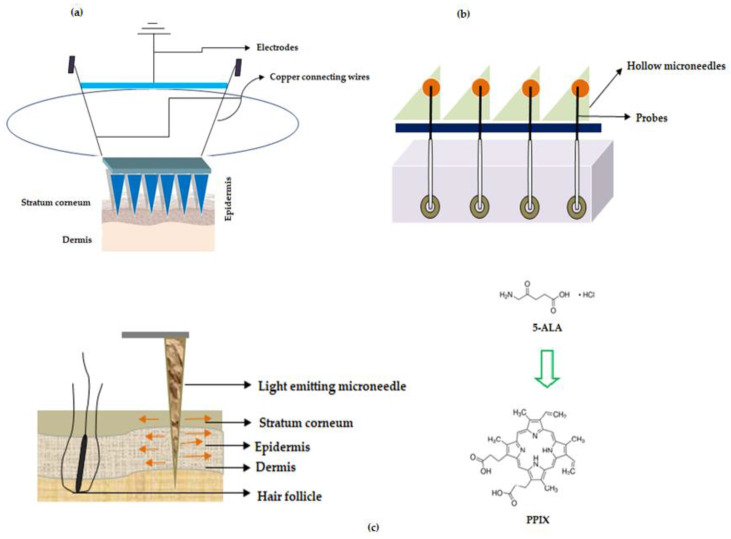
Microneedles for treatment of cancer (**a**) Illustration showing MNs with sensor unit when applied to skin (**b**) Microneedle array device with probe (**c**) Light emitting fiber optic microneedle converting 5-Aminolevulinic acid (5-ALA) to Protoporphyrin IX (PPIX).

**Figure 7 pharmaceutics-12-01101-f007:**
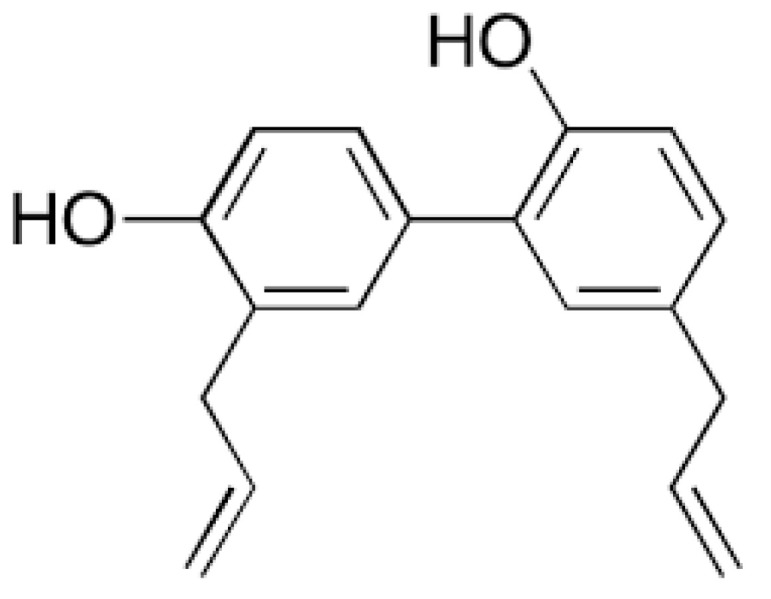
*Honokiol:* Biphenyl natural anti-cancer compound from *Magnolia glandiflora.*

**Figure 8 pharmaceutics-12-01101-f008:**
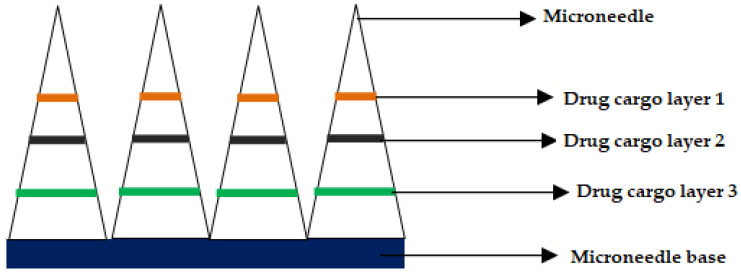
Cross-section of the microneedle after drug cargo loading by micro-milling.

**Figure 9 pharmaceutics-12-01101-f009:**
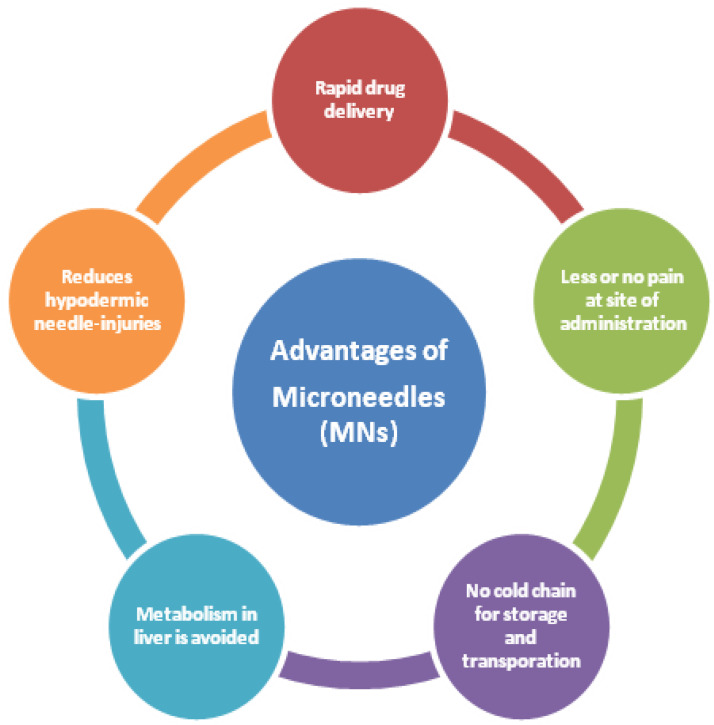
Advantages of MNs.

**Table 1 pharmaceutics-12-01101-t001:** Different types of analytes relied on to detect cancer using microneedles.

Name of the Analyte	Principle Involved in Detection	Structure and Materials for Sensor	Test Subject	Ref.
Glucose	Colorimetry Glucose assay Electrochemistry	Hydrogel solid MNs Metal hollow MNs and paper sensor Silicon hollow MNs and sensor	Mouse Rabbit Human	[117,118,119]
Glutamate	Electrochemistry	Hollow or solid MNs of a polymer	-	[120]
Lactate	Electrochemistry	Hollow or solid MNs of polymer along with carbon paste or carbon nanotubes	-	[121,122]
Hydrogen peroxide	Electrochemistry	Solid MNs with platinum or gold electrodes	Mouse	[123]
Nitric oxide	Electrochemistry	Solid MNs of a polymer or metal with hemin or graphene that is functionalized.	Melanoma mouse Rat	[124,125]
T cell	Immune response	Solid MNs of polymer with antigens (nano-capsule)	Human skin	[126]
Potassium	Electrochemistry	Hollow MNs of polymer or metal with ion specific electrode	Porcine skin	[127,128]

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
