# Peer review of "Microneedles Drug Delivery Systems for Treatment of Cancer: A Recent Update"

_pharmaceutics, 2020, doi:10.3390/pharmaceutics12111101_

Round 1

Reviewer 1 Report

Pharmaceutics 986714

Microneedles Drug Delivery Systems for Treatment of Cancer: A Recent Update

Aravindram Attiguppe Seetharam1, Hani Choudhry2, Muhammed A Bakhrebah3, Wesam H Abdulaal2, Maram Suresh Gupta1, Syed Mohd Danish Rizvi4, Qamre Alam5, Siddaramaiah6, and Devegowda Vishkante Gowda1* and Afrasim Moin4**  

1Department of Pharmaceutics, JSS College of Pharmacy, JSS Academy of Higher Education and Research  (JSSAHER), Sri Shivarathreeshwara Nagar, Mysore, 570 015, India.

2Department of Biochemistry, Cancer Metabolism & Epigenetic Unit, Faculty of Science, Cancer & Mutagenesis Unit, King Fahd Medical Research Center, King Abdulaziz University, Jeddah 21589, Saudi Arabia.

3Life Science & environment Research Institute, King Abdulaziz City for Science and Technology (KACST), Riyadh 11442, Saudi Arabia.

4Department of Pharmaceutics, College of Pharmacy, University of Hail, Hail, Kingdom of Saudi Arabia.

5Medical Genomics Research Department, King Abdullah International Medical Research Center (KAIMRC), King Saud Bin Abdulaziz University for Health Sciences, King Abdulaziz Medical City, Ministry of National Guard Health Affairs, Riyadh, Kingdom of Saudi Arabia.

6 Department of Polymer Science and Technology, Sri Jayachamarajendra College of Engineering, Mysore, 570 17 016, India.

The authors present a review of microneedle technologies for the treatment of cancer.

Major comments:

  • The authors justify the value of the present review in lines 76-87, and line 113-114. It is the opinion of this reviewer that the value of an article should be obvious to the reader and not require advertisement within the article itself. Inclusion of such a section is of poor form and should be removed.
  • Section 2.4.3 provides specific examples where microneedle technology can improve therapeutic outcome. However, a more general discussion of this topic should be included in addition to these specific examples. Some questions that remain unaddressed include: Which drug types and doses are suitable for delivery using microneedles? What are the advantages and limitations of transdermal drug delivery using microneedles? Differences between local and systemic drug delivery using microneedles?
  • The authors should provide their expert perspective with regard to the prospects of microneedle technology for the diagnosis and treatment of cancers.
  • Section 3.1. It is not immediately obvious how microneedle sensor technologies are relevant for the treatment of cancer. This should be made clear at the beginning of section 3 or 3.1.
  • Section 3.4. should either be expanded to provide a more thorough discussion, or included within section 2.3 as an alternative manufacturing method.
  • The structure, text, and tone must be drastically improved.

Author Response

Dear Honorable Reviewer

Thanks & Regards

Reviewer 2 Report

This is a review article intended to provide an overview of the implementation of microneedle based technologies to deliver therapeutics for the treatment of cancer.

However, 40-50% of this article goes into the general background of the microneedle such as types of microneedle, different geometries, materials used, clinical trials, and outcomes. Entire section 2 is dedicated to the aforementioned topics it must be noted that these subject matters have been very well established in the literature and this section does not add any value to the article. The authors would be well advised that instead of giving a general overview, they should focus on how these aspects of microneedle technology have evolved to improve the treatment of cancer. If this section cannot be made specific to cancer, it is recommended that this entire section should be removed.

The authors should remove Section 6 completely as this draft guidance by FDA does not include the combination of drugs with the device. Typically, they are governed by regulations covered under "Combination Products" guidelines and the current draft clearly states that combination products are not under the scope of the document. 

Section 3 is which provides a review of different drugs and microneedle technologies used for cancer treatment, is crafted well. This section can be broken up into two sections (after removing section 2 and section 6). One section can focus on different cancer treatments using microneedles and the other section can focus on more recent manufacturing methods (like 3D printing and additive manufacturing). 

Author Response

(The authors gave the same response as above.)

Reviewer 3 Report

 Figure quality was improved compared to previous version. Authors performed revision appropriately. This paper is suitable for publication in this journal.

Round 2

Reviewer 1 Report

The authors have addressed the reviewer's comments.

Reviewer 2 Report

Two out of three major comments have been addressed and I find the changes suitable.

The authors have provided a reasonable rebuttal to keep section 2. I would still suggest tailoring that section more towards cancer treatment if that is feasible.